# Use of a Cross-Sectional Survey in the Adult Population to Characterize Persons at High-Risk for Chronic Obstructive Pulmonary Disease

**DOI:** 10.3390/healthcare7010012

**Published:** 2019-01-18

**Authors:** Roy A. Pleasants, Khosrow Heidari, Jill Ohar, James F. Donohue, Njira Lugogo, Chelsea L. Richard, Sarojina Kanotra, David M. Mannino, Monica Kraft, Winston Liao, Charlie Strange

**Affiliations:** 1Division of Pulmonary Diseases and Critical Care Medicine, Department of Medicine, University of North Carolina at Chapel Hill, Chapel Hill, NC 27599, USA; james_donohue@med.unc.edu; 2Durham VA Medical Center, Durham, NC 27708, USA; 3South Carolina Department of Health and Environmental Control, Columbia, SC 29201, USA; heidarik@dhec.sc.gov (K.H.); richarcl@dhec.sc.gov (C.L.R.); 4Pulmonary, Critical Care, Allergy, Immunologic Diseases Section, Wake Forest University, Winston-Salem, NC 27109, USA; johar@wakehealth.edu; 5Division of Pulmonary and Critical Care Medicine, University of Michigan, Ann Arbor, MI 48109, USA; nlugogo@med.umich.edu; 6Kentucky Department for Public Health, State Public Health Agency, Frankfort, KY 40621, USA; sarojini.kanotra@ky.gov; 7Department of Medicine, University of Kentucky, Lexington, KY 40506, USA; dmmann2@email.uky.edu; 8GlaxoSmithKline PLC, Philadelphia, PA 19112, USA; 9Department of Internal Medicine, University of Arizona, Tucson, AZ 85721, USA; mkraft@deptofmed.arizona.edu; 10Independent Researcher, Chapel Hill, NC 27599, USA; winston.liao1@gmail.com; 11Division of Pulmonary, Critical Care and Sleep Medicine, Medical University of South Carolina, Charleston, SC 29425, USA; strangec@musc.edu

**Keywords:** productive cough, shortness of breath, dyspnea on exertion, health impairment, comorbidities, tobacco smoking, body mass index, undiagnosed COPD

## Abstract

Rationale/Objective: The Behavioral Risk Factor Surveillance System (BRFSS) health survey has been used to describe the epidemiology of chronic obstructive pulmonary disease (COPD) in the US. Through addressing respiratory symptoms and tobacco use, it could also be used to characterize COPD risk. Methods: Four US states added questions to the 2015 BRFSS regarding productive cough, shortness of breath, dyspnea on exertion, and tobacco duration. We determined COPD risk categories: provider-diagnosed COPD as self-report, high-risk for COPD as ≥10 years tobacco smoking and at least one significant respiratory symptom, and low risk was neither diagnosed COPD nor high risk. Disease burden was defined by respiratory symptoms and health impairments. Data were analyzed using multiple logistic regression models with age as a covariate. Results: Among 35,722 adults ≥18 years, the overall prevalence of COPD and high-risk for COPD were 6.6% and 5.1%. Differences among COPD risk groups were evident based on gender, race, age, geography, tobacco use, health impairments, and respiratory symptoms. Risk for disease was seen early where 3.75% of 25–34 years-old met high-risk criteria. Longer tobacco duration was associated with an increased prevalence of COPD, particularly >20 years. Seventy-nine percent of persons ≥45 years-old with frequent shortness of breath (SOB) reported having or being at risk of COPD, reflecting disease burden. Conclusion: These data, representing nearly 18% of US adults, indicates those at high risk for COPD share many, but not all of the characteristics of persons diagnosed with the disease and demonstrates the value of the BRFSS as a tool to define lung health at a population level.

## 1. Introduction

Characterizing COPD at the population level through cross-sectional surveys is important to the overall strategy to target the disease. Both the National Heart Lung and Blood Institute’s (NHLBI) National COPD Action Plan [1] and the Center for Disease Control and Prevention’s (CDC) Public Health Strategic Framework for COPD Prevention [2] recommend the expansion of epidemiologic instruments in the USA to better characterize persons with and at risk for disease. One goal of the National COPD Action Plan is to “enhance and optimize capacity” to collect epidemiologic data from multiple sources, at various geographic levels, and to act upon these data.

The CDC has several surveillance tools that could be used to characterize the burden of COPD in the population, in some instances in specific geographic regions as recommended by the NHLBI. The nationally-based, state-administered BRFSS telephone health survey, conducted annually in >400,000 adults, has been used to estimate COPD prevalence since 2007 [3]. Whereas the core BRFSS includes questions regarding tobacco use as smoking status, it does not define tobacco intensity or duration, key determinants of tobacco-related health risks [4]. In 2012, we used the BRFSS in one state to better characterize persons with or at risk for COPD through the inclusion of tobacco smoking duration and respiratory symptom questions [5]. To explore this approach in a larger population, we used these same survey questions in 2015 to characterize adults at high-risk of COPD in four states, that collectively represent 17.7% of the US adult population [6].

## 2. Methods

### 2.1. Study Population and Survey

State health departments collaborate with the Centers for Disease Control and Prevention (CDC) to conduct the BRFSS. It is a random, digit-dialed cellphone and landline survey of non-institutionalized adults ≥18 years, composed of a core section of >90 questions about socio-demographics, health risk behaviors and characteristics, chronic diseases and preventive health practices. Individual states can add supplementary questions. We added a module of four questions of the type and frequency of respiratory symptoms and tobacco smoking duration in four state’s BRFSS in 2015. Florida (FL) and Texas (TX) were selected due to a high number of reported COPD hospitalizations, Kentucky (KY) because of high COPD prevalence, and South Carolina (SC) to assess survey replicability compared to 2012 data. Combined, an estimated 18.7% of COPD patients reside in these states [7].

The BRFSS is approved as exempt research by the CDC. Survey response rates (proportion of all eligible and likely-eligible persons) were calculated using standards by the American Association of Public Opinion Research Response Rate Formula #4 (https://www.aapor.org/AAPOR_Main/media/publications/Standard-Definitions2015_8theditionwithchanges_April2015_logo.pdf). The response rates for each state were as follows—landline, cellphone, and combined: FL 37.0%, 37.1%, and 37.0%; KY 62.1%, 51.6%, and 59.0%; SC 52.1%, 51.5%, and 50.4%; TX 32.9%, 41.0%, and 34.4%. The number of respondents were—FL (9739), KY (8806), SC (11,607), and TX (14,697). Detailed information is available through the 2015 BRFSS Summary Data Quality Report at CDC BRFSS (http://www.cdc.gov/surveillancepractice/reports/brfss/brfss.html).

Module questions were derived from COPD screening questionnaires [8,9,10,11] (Table 1) and are the same as those added to the 2012 SC BRFSS [5]. To help differentiate between low and high-risk for COPD, we did not include intermediate respiratory symptoms in the analysis such as productive cough a few days each month. When shortness of breath (SOB) was reported as slightly or strongly associated with physical activities, it was considered to represent dyspnea on exertion (DOE). Those with a positive smoking history were asked about life-time duration of tobacco smoking (tobacco-years). Those with a positive smoking history were defined as those respondents answering affirmatively to “Have you smoked at least 100 cigarettes in your entire life?” Those responding “no” were categorized as never smokers (Table 1). Individuals responding “yes” were also asked, “Do you now smoke cigarettes every day, some days, or not at all?” Persons who responded “not at all” were defined as former smokers with the remainder defined as current smokers.

### 2.2. COPD Risk Groups

Among persons ≥18 years-old, three COPD risk groups were determined based upon self-report of diagnosis, respiratory symptoms, and tobacco-years. Diagnosed COPD was a self-report of provider-diagnosed disease; high-risk for COPD were respondents with no COPD diagnosis, ≥10 tobacco-years, and ≥1 significant respiratory symptom (frequent productive cough, frequent SOB, or DOE); and low-risk as neither diagnosed COPD nor at high-risk.

### 2.3. Disease Burden

Questions and affirmative responses for health impairment are shown in Table 1. Other measures of disease burden included comorbidity frequencies. The BRFSS includes questions regarding comorbidities as depression, asthma, coronary heart disease, and arthritic among others based on self-report of provider diagnosis. Body mass index (BMI) was determined from reported height and weight without shoes and categorized using World Health Organization classifications [12].

### 2.4. Analysis

Respondents were excluded from analysis for missing data on COPD history, smoking status, smoking duration, or one or more respiratory symptoms. All analyses were conducted using SAS 9.4 to account for the complex BRFSS sampling. Analysis was done collectively for all states and for selected measures by individual state. First, we calculated the percentage and 95% confidence intervals (CI) of socio-demographics, health behaviors, health impairment, and chronic diseases. For comparisons of prevalence’s between subgroups, statistical significance was determined by t-tests. We tested for linear trends across categories (e.g., tobacco duration and age) using the Wald F-statistic. Age-adjusted prevalence and 95% CI of selected characteristics by COPD risk status, age, tobacco duration, was calculated. Prevalence was compared by risk status using pairwise *t*-tests. A *p*-value <0.05 was considered statistically significant for all tests.

## 3. Results

A total of 44,849 subjects ≥18 years-old were asked core BRFSS questions including COPD prevalence. The module of four questions to determine COPD risk was asked of all states’ respondents over the 12-month period except FL where this was asked over 6 months. After exclusion of subjects with incomplete data there were 35,722 subjects’ data analyzed. More than half of the respondents were ≥45 years-old, female, Caucasian, and at least some college education. Hispanic and African-American minorities were well represented (Table 2).

### 3.1. COPD Risk

Table 2 shows the socio-demographics, smoking status, and tobacco-years among the COPD risk groups. Among adults over 18 years-old, the overall self-reported, provider-diagnosed COPD prevalence was 6.6% and high-risk for COPD was 5.1%. Women reported higher age-adjusted prevalence of COPD (*p* < 0.05), whereas men were more likely to be at high-risk for COPD (*p* < 0.05). Caucasians and multi-racial persons had the highest rates of provider-diagnosed COPD and Hispanics the lowest (*p* < 0.05); Caucasians also reported the highest prevalence of high-risk for COPD. Those with <high school education had higher rates of COPD and at high-risk for COPD than those with a college education (*p* < 0.05). Figure 1 shows the frequencies of COPD, at-high risk for COPD, respiratory symptoms and current smoking status for each age group. The prevalence of high-risk was 3.75% in the 25–34 years age group, further increased with age and then remained relatively stable after 44 years-old. After DOE, frequent productive cough was the next most common symptom (Figure 1). The increase in COPD prevalence was most evident between the 45–54 years-old and 54–65 years-old age groups. Combined, nearly 60% of persons at high-risk for COPD were in the 45–54 and 55–64 years-old age groups.

### 3.2. Tobacco Smoking

The frequencies of high-risk for COPD and self-report of provider-diagnosed COPD, in relationship to tobacco status and tobacco-years are shown in Table 2. 72.6% (95% CI 71.4–73.7) of persons in the low risk group were never smokers and 9.3% (95% CI 8.5–10.1) smoked <10 years. Figure 1 shows current smoking rates among age groups for the four states combined and Figure 2 shows data for individual states. The risk of self-reported COPD was most apparent with >20 years of smoking, with rates exceeding 30% in those with >30 years. Current smoking rates peaked in the 25–34 years-old age group, remained somewhat stable until the 55–64 years-old age group, declining thereafter (Figure 1 and Figure 2).

### 3.3. Comorbidities

Comorbid conditions among COPD groups are shown in Table 3. All comorbidities occurred more often in high-risk for COPD and COPD groups compared to low-risk, except cancers. The most common chronic illnesses in the high-risk and COPD groups were depression and arthritis, followed by asthma. Some differences in frequency of comorbidities between high-risk and COPD groups were found. Concurrent asthma was reported in nearly 10% of persons at high-risk of COPD and by more than a third of COPD patients—two to four-fold greater than the low-risk group.

### 3.4. Health Impairment

Among health impairment measures, fair or poor overall health status and difficulty walking or climbing stairs were less frequent in the high-risk group than the COPD group (*p* < 0.05); physical and mental distress were similar between these two groups. All health impairments were less frequent in the low-risk group (*p* < 0.05). One-third of persons at high-risk for COPD and COPD reported frequent productive cough, and was greater than the low-risk group (*p* < 0.05). Frequent SOB was the only respiratory symptom significantly less common in those at high-risk versus COPD (*p* < 0.05).

### 3.5. State-Specific Data

The prevalence of high-risk for COPD, self-reported, provider-diagnosed COPD, and low-risk for COPD from each state are shown in Table 4. Rates of high-risk for COPD and COPD ranged from 3.7–9.0% and 5.7–11.2%, respectively, and were highest in KY and lowest in TX, consistent with the current-smoker rates for each state shown in Figure 2. Frequent SOB was reported in adults > 18 years-old (%, 95% CI) as follows: FL (3.4%, 2.4–4.4), KY (6.8%, 5.6–7.9), SC (3.8%, 3.3–4.3), and TX (2.3%, 1.9–2.7). Among survey respondents ≥45 years-old in these states who reported frequent SOB, nearly 80% of adults also reported either being diagnosed with COPD or were defined as high-risk for COPD, also varying by state (Table 5).

## 4. Discussion

We used a cross-sectional, population-based health survey in four states to better characterize persons at significant risk of COPD consistent with expanding disease surveillance recommendations by the NHLBI and CDC. In a study of four states representing 17.7% of the US adult population, using frequent respiratory symptoms and >10 years tobacco smoking history as risk factors, 5.1% of adults were considered to be at high-risk for COPD and thus possibly undiagnosed or pre-disposed to developing disease. A new finding from this study is that adults ≥45 years-old at high-risk of COPD or reported having COPD accounted for 79.2% of adults who report frequent SOB, indicative of the substantial disease burden at the population level. We also show that early COPD becomes evident in young adults where 2.4% of 18–24 years-old reported provider-diagnosed COPD and 3.75% of 25–34 years-old met the definition of being at high-risk. Further, this study demonstrates that prolonged tobacco duration (tobacco-years) increases the likelihood of being at high-risk for COPD as well as corroborates some less well-established findings as the association of tobacco duration and increased risk of diagnosed COPD [13].

The BRFSS health survey is conducted annually in states throughout the USA and its territories, thus incorporating respiratory symptoms and tobacco duration questions could be used to better describe and monitor general lung health at different geographic levels. Strengths of the survey include representation of the individual’s perspective of symptoms, behaviors and health impairments not typically available in healthcare system databases. Respiratory symptom questions employed for this study were derived from multiple questionnaires used to screen for COPD [8,9,10,11], and these characterize the most common pulmonary symptoms associated with COPD—mucus production and dyspnea—as well as the main risk factor for lung disease in the US—prolonged tobacco smoking [14]. Whereas the core BRFSS questions address a number of tobacco-related measures as status and cessation attempts, it does not define tobacco smoking intensity nor duration, key factors affecting tobacco-related morbidity and mortality [4]. Tobacco-years is a valid measure of risk, as duration is a stronger predictor than intensity (pack-years) for risk of lung cancer [15], heart disease [16], as well as COPD and emphysema [13]. Therefore, inclusion of tobacco duration is an important contribution to the BRFSS.

Based on symptoms and tobacco smoking history, nearly as many adults >18 years-old were at high-risk for COPD as there were persons reporting having COPD. In prior studies of persons with undiagnosed COPD, airflow obstruction was most often mild, however, more severe disease was also found [17,18]. Defining early or undiagnosed COPD can lead to higher smoking cessation rates [19], but other benefits as increased survival are yet to be established, although some evidence shows treatment in early, mild COPD slows decline in lung function [20,21]. The United States Preventive Services Taskforce recommends the use of spirometry to screen for COPD in symptomatic persons [22]. As COPD is a leading cause of death in the USA, one could argue the best approach would be an aggressive, comprehensive strategy to screen and diagnose disease. Studies as the RETHINC trial [23] are evaluating benefits of pharmacological treatment of early disease.

There is increasing emphasis on targeting persons with early COPD, those at risk of developing disease including young adults, as well as persons with COPD but are undiagnosed [1,2,24]. Using the criteria of a minimum of 10 tobacco-years and at least one significant respiratory symptom, high-risk for COPD was evident as early as 25 years-old (2.5%), the rate then increased, and plateaued at 45 years-old. This pattern could reflect persons eventually being diagnosed in their 40′s and 50′s, their disease did not progress to COPD [25] or early mortality [26,27]. Smoking, environmental exposures, and early life lung function are important determinants for lung function decline [25,28,29]. Smoking rates were highest in the 25–34 years-old, likely a contributor to the respiratory symptoms as productive cough in this age group and thus a self-report of a provider-diagnosis of COPD. Asthma early in life appears to be an important risk factor for developing COPD, especially in smokers [25,28,29]. We found that asthma was more common in the high-risk group than the low-risk group, and the greatest prevalence was in those with diagnosed COPD.

Martinez et al. proposed that early changes leading to COPD should be studied in persons <50 years-old with ≥10 pack-years using objective measures as spirometry and radiography to define early disease, but not symptoms [24]. They did point out the risk of COPD is greater in symptomatic persons without airflow obstruction, however there was a dissociation between structural lung damage and symptoms. Although it is unknown what portion of subjects in our study who reported frequent respiratory symptoms had AO by spirometry or structural changes, the presence of symptoms are invariably important and are known to affect overall well-being and long-term outcomes [27,30,31,32,33,34,35,36,37], and may be a better predictor of mortality than forced expiratory volume in 1 s (FEV_1_) [27,36,38]. In the COPDGene study, symptomatic ever-smokers without airflow obstruction, regardless of history of asthma, had greater activity limitation, lower pulmonary function, and more airway-wall thickening by computed tomography than did asymptomatic ever-smokers [37].

Health impairments, respiratory symptoms, and comorbidities were common in persons at high-risk for COPD as has been reported persons with undiagnosed COPD (31) as well as in those with significant respiratory symptoms without airflow obstruction [37,39]. In our study, where those at high-risk likely include undiagnosed persons, measures of physical impairment were evident by high rates of DOE and difficulty in walking or climbing stairs. Physical impediments contribute to deconditioning and worsened dyspnea [40], whereas mental health issues as anxiety and depression contribute to increased respiratory symptoms [41]. Mental health effects were apparent in the high-risk group as shown by high rates of depression and frequent mental distress, similar to that reported by those with COPD.

The effects of gender on COPD risk were found in this study as men were more likely to be at high-risk for COPD, and conversely women were more likely to be diagnosed with COPD. Although women are less likely to smoke and have shorter lifetime tobacco duration than men, they are more susceptible to the effects of tobacco [42] and use a greater portion of their lung capacity [43]. Further, as women are more likely to report dyspnea than men [44], they may be more likely to be diagnosed with COPD, yielding a higher disease prevalence [7] and ultimately greater associated mortality [45]. The under recognition of COPD in men represents an important health disparity warranting greater emphasis on early diagnosis.

Consistent with our study, racial differences in COPD prevalence have previously been reported where Caucasians have higher frequencies than Hispanics and African Americans [5,7,46]. The lower rate of either being at high-risk for COPD or having COPD found in Hispanics and African Americans may be related to differences in tobacco smoking, genetics, and potentially other factors, such as healthcare access. Whereas African Americans have similar ever-smoking rates compared to Caucasians, they have lower tobacco intensity (pack years) [47]. Compared to Caucasians, Hispanics have lower rates of ever smoking [4]. In the current study, racial differences in COPD risk appeared to contribute to geographic variation among states. The 2010 US census reported the Hispanic census in the four states participating in this study as FL 26%, KY 5%, SC 5%, and TX 38% [6], consistent with the relative prevalence of high-risk for COPD and diagnosed COPD in each state. Whereas we found a lower prevalence of being at high-risk of COPD in African Americans than Caucasians, one study has reported higher rates of undiagnosed COPD in African Americans, but that study was not cross-sectional and racial differences were therefore dependent on study enrollment [48].

Tobacco smoking is the principal risk factor for COPD in the USA—we found that nearly one-half of the study participants were ever-smokers and rates varied among states and with age. Although we did not measure intensity, we found prolonged duration was associated with an increased likelihood of being at high-risk for COPD as well as being diagnosed with COPD. The 25–34 years-old age group represents an important target for smoking cessation interventions. We found COPD risk increased significantly somewhere between 10 and 20 years of smoking, and especially ≥30 tobacco-years. As smokers vary in their tobacco use over time, it is possible that a smoking history based on years is more reliable than one based on years and number of packs/day or cigarettes smoked/day. In addition, quantifying cigar and pipe smoking intensity is difficult based on pack years. Considering the long-held use of pack years to define smoking history in the clinical setting and some of its potential inaccuracies, reporting both pack years and tobacco-years in medical records may be a rational approach. More research needs to be undertaken to evaluate severity of disease as well as risk of developing COPD in relationship to tobacco duration and intensity, and therefore the best approach to define and document tobacco smoking use.

### Limitations

Our study had several limitations including those inherent to the BRFSS, as responses were obtained by telephone survey; therefore, we relied on self-report of medical conditions and smoking history. COPD diagnoses were not confirmed with spirometry or review of medical records, which could result in misclassification of respondents, however, previously, three-fourths of BRFSS respondents with COPD reported having a diagnostic breathing test [49]. When verified by spirometry, sensitivity has ranged from 78% to 99% for a self-report of COPD [50,51,52]. We did not define DOE or SOB using standard dyspnea questions, although one could argue SOB affecting physical activities has more value than SOB walking on an incline. Although we defined cough over the last month whereas GOLD definition uses two-year period symptoms [14], the St George Respiratory Questionnaire uses a one-month period to define productive cough [53]. Lastly, we did not address other non-tobacco causes of COPD as occupation or biomass exposures, which may underestimate prevalence of being at high-risk for COPD.

## 5. Conclusions

From a large recent cross-sectional respiratory symptom health survey conducted in adults in four USA states, we provide useful insight into prevalence and burden of disease in persons at high-risk of COPD. Whereas self-report of provider-diagnosed COPD prevalence was higher in women, men were more likely to be characterized as high-risk for COPD. The frequencies of comorbidities and health impairment measures were similar between COPD and high-risk for COPD groups. Among the three respiratory symptoms measured in this study, dyspnea was the only symptom more common in COPD than at high-risk of COPD. There was wide variation in COPD risk among the states and among different races. Adults in the four states with COPD or at high-risk of COPD accounted for more than three-fourths of persons in the population reporting frequent SOB. The BRFSS health survey could be used as a tool to monitor national respiratory health in a large population and aid initiatives to improve lung diseases by characterizing at-risk individuals and thus help support national health strategies for COPD, such as the NHLBI National COPD Plan.

## Figures and Tables

**Figure 1 healthcare-07-00012-f001:**
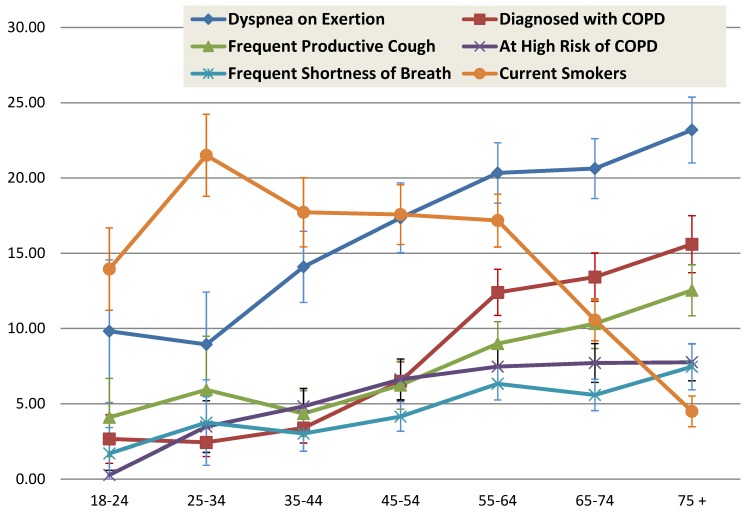
Age group specific prevalence of respiratory symptoms, current smoking, and COPD risk in adults ≥18 years-old using the 2015 Behavioral Risk Factor Surveillance System telephone survey in four states (FL, KY, SC, TX).

**Figure 2 healthcare-07-00012-f002:**
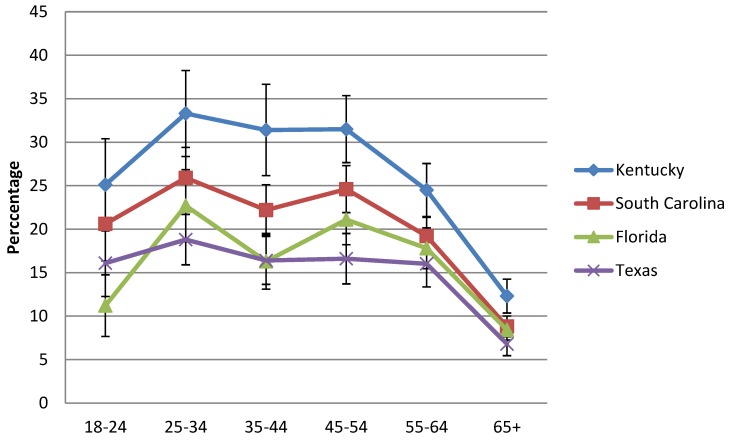
Current smoking rates based on age group (years) for FL, KY, SC, and TX using the 2015 Behavioral Risk Factor Surveillance System health survey.

**Table 1 healthcare-07-00012-t001:** Respiratory module and health impairment questions with possible responses using the 2015 Behavioral Risk Factor Surveillance System telephone health survey in four states (FL, KY, SC, TX).

**Respiratory Symptom Module**	**Definition of Affirmative Response**
How many years have you smoked tobacco products?___ yrs (Above question asked if respondent answered affirmatively to the following—Have you smoked more than 100 cigarettes in your lifetime?	≥10 years of tobacco smoking (10 tobacco-years) considered risk factor for developing COPD.
During the past 30 days, how often do you feel short of breath?—would you say all of the time, most of the time, some of the time, a little of the time, or none of the time?	Frequent SOB defined as all of the time or most of the time.
During the past 30 days, how often do you cough up mucus or phlegm? Would you say…? Every day, Most days, A few days, Only with colds, or Never.	Frequent productive cough defined as every day or most days.
How much do you agree or disagree with the following statement? In the past year, I am not as physically active as I once was because of my shortness of breath. Agree strongly, Agree slightly, Neither agree or disagree, Disagree slightly, Disagree strongly.	Dyspnea on exertion defined as agreeing strongly or slightly.
**Health Impairment**	**Definition of Affirmative Response**
Would you say your general health is… excellent, very good, good, fair, or poor?	A reply of fair or poor.
Now thinking about your mental health, which includes stress, depression, and problems with emotions, for how many days during the past 30 days was your mental health not good?	Frequent mental distress was defined as a response of ≥14 days to the question.
Now thinking about your physical health, which includes physical illness and injury, for how many days during the past 30 days was your physical health not good?	Frequent physical distress was defined as a response of ≥14 days to the question.
Do you have serious difficulty walking or climbing stairs?	A reply of yes.

**Table 2 healthcare-07-00012-t002:** Age adjusted prevalence and 95% confidence intervals (CI) of selected characteristics adults aged ≥18 years by COPD risk status ^a–c^ using BRFSS, 2015 among Four States (FL, KY, SC, TX).

Characteristic			DN with COPD ^a^ (n = 3915) (%, 95% CI)		HR for COPD ^b^ (n = 2399) (%, 95% CI)		LR for COPD ^c^ (n = 29,585) (%, 95% CI)		Contrast (*P*-Value)		
									DN vs. HR	DN vs. LR	HR vs. LR
Mean Age (Years, 95% CI)			59.0	(57.7–62.0)	55.2	(53.7–56.8)	46.4	(46.0–46.8)	< 0.001	<0.001	<0.001
Overall			6.6	(6.1–7.1)	5.1	(4.6–5.6)	88.2	(87.6–89.0)	<0.001	<0.001	<0.001
Socio-demographics	Sex	Men	6.2	(5.4–6.9)	6.3	(5.5–7.1)	87.6	(86.5–88.6)	NS	<0.001	<0.001
		Women	7.0	(6.4–7.7)	4.1	(3.5–4.9)	88.8	(87.9–89.7)	<0.001	<0.001	<0.001
	Age (years) (Not age-adjusted)	18–25	2.7	(1.1–4.3)	0.3	(0.0–0.6)	97.1	(95.4–98.7)	<0.001	<0.001	<0.001
		25–34	2.4	(1.5–3.4)	3.5	(1.8–5.2)	84.1	(92.2–96.0)	<0.001	<0.001	<0.001
		35–44	3.4	(2.7–4.7)	4.8	(3.6–6.0)	91.8	(90.3–93.3)	NS	<0.001	<0.001
		45–54	6.5	(6.1–8.7)	6.6	(5.2–8.0)	86.9	(85.1–88.7)	NS	<0.001	<0.001
		55–64	12.4	(10.9–13.8)	7.4	(6.3–8.6)	80.1	(78.3–82.0)	<0.001	<0.001	<0.001
		65–74	13.4	(11.9–14.9)	7.7	(6.4–9.0)	78.9	(77.0–80.8)	<0.001	<0.001	<0.001
		≥75	15.6	(13.7–17.5)	7.8	(6.5–9.0)	76.6	(74.5–78.8)	<0.001	<0.001	<0.001
	Race	White—Non-Hispanic	7.8	(7.1–8.5)	6.5	(5.6–7.3)	85.7	(84.6–86.8)	<0.001	<0.001	<0.001
		Black—Non-Hispanic	5.4	(4.0–6.8)	3.5	(2.6–4.4)	91.1	(89.5–92.7)	NS	<0.001	<0.001
		Hispanic	4.2	(3.1–5.3)	3.3	(2.4–4.2)	92.5	(91.1–93.9)	NS	<0.001	<0.001
		Other race only, Non-Hispanic	8.6	(3.4–9.8)	5.6	(2.8–8.4)	87.8	(83.7–91.9)	NS	<0.001	<0.001
		Multiracial, Non-Hispanic	11.2	(6.5–15.9)	8.7	(2.0–15.4)	80.1	(72.3–88.0)	NS	<0.001	<0.001
	Education	< High School	9.1	(7.8–10.5)	6.6	(5.6–9.7)	83.3	(80.9–85.7)	NS	<0.001	<0.001
		High School Graduate	7.8	(6.8–8.8)	6.4	(5.3–7.5)	85.7	(84.3–87.2)	NS	<0.001	<0.001
		Some College	6.6	(5.6–7.5)	5.2	(4.3–6.1)	88.3	(87.0–89.5)	<0.001	<0.001	<0.001
		College Graduate	3.6	(2.9–4.2)	2.3	(1.8–2.7)	94.2	(93.4–94.9)	<0.001	<0.001	<0.001
Risk factors	Smoking status	Current smoker	18.2	(15.9–20.4)	21.4	(18.5–24.2)	60.5	(57.2–63.7)	NS	<0.001	<0.001
		Former smoker	8.2	(7.1–9.4)	8.5	(7.0–10.1)	83.2	(81.4–85.19)	NS	<0.001	<0.001
		Never smoker	3.0	(2.6–3.5)	~	~	96.9	(95.5–97.4)	<0.001	<0.001	<0.001
	Tobacco smoking duration (years) (Not age-adjusted)	Never smoked	3.0	(2.6–3.5)	~	~	97.0	(96.5–97.4)	<0.001	<0.001	<0.001
		1–9	5.0	(3.7–6.4)	~	~	95.0	(93.6–96.3)	<0.001	<0.001	<0.001
		10–19	8.4	(6.4–10.4)	23.5	(19.9–27.1)	68.1	(63.3–71.8)	<0.001	<0.001	<0.001
		20–29	16.8	(14.1–18.4)	21.8	(18.9–24.6)	61.5	(58.0–64.9)	<0.001	<0.001	<0.001
		≥30	31.1	(28.3–33.9)	23.9	(21.2–26.6)	45.0	(41.9–64.9)	<0.001	<0.001	<0.001

DN: Diagnosed; HR: High-Risk; LR: Low-Risk. ^a^ Diagnosed-Self-reported physician diagnosis of COPD; ^b^ High-risk-No COPD diagnosis, ≥10 years of cigarette/tobacco use, and at least one respiratory symptom (frequent productive cough, frequent shortness of breath (SOB), or breathing problems limited physical activity in past year.); ^c^ Low-risk—Neither diagnosed with nor high-risk for COPD.

**Table 3 healthcare-07-00012-t003:** Age adjusted prevalence and 95% confidence intervals (95% CI) of health characteristics and respiratory symptoms in adults aged ≥18 years by COPD risk status ^a^—BRFSS, 2015 among four states (FL, KY, SC, TX).

Characteristic	DN with COPD (n = 3915) (%, 95% CI)		HR for COPD (n = 2399) (%, 95% CI)		LR for COPD (n = 29,585) (%, 95% CI)		Comparison of Groups (*P*-Value)		
							DN vs. HR	DN vs. LR	HR vs. LR
**Chronic Conditions**									
Current asthma	34.6	(29.7–39.4)	8.7	(5.7–11.6)	4.6	(4.1–5.2)	<0.001	<0.001	<0.001
Coronary heart disease	15.0	(12.2–17.7)	10.4	(7.9–12.9)	3.5	(3.1–3.9)	NS	<0.001	<0.001
Diabetes mellitus	22.4	(18.8–25.9)	17.1	(13.9–20.2)	13.0	(12.1–13.9)	NS	<0.001	<0.001
Arthritis	59.8	(54.1–65.6)	42.0	(37.0–47-0)	25.7	(24.6–26.8)	<0.001	<0.001	<0.001
Cancer (excluding skin)	7.8	(4.9–10.8)	4.1	(2.7–5.3)	3.5	(3.0–3.9)	NS	<0.001	NS
Depression	48.2	(42.6–53.9)	33.1	(28.0–38.2)	13.2	(12.2–14.2)	<0.001	<0.001	<0.001
Stroke	11.7	(9.0–14.4)	6.0	(4.3–7.6)	2.8	(2.4–3.1)	<0.001	<0.001	<0.001
Kidney disease	10.3	(5.7–14.9)	4.5	(3.0–5.9)	2.7	(2.3–3.1)	NS	<0.001	NS
**Health Impairment**										
Fair/poor health ^b^	53.5	(48.1–60.0	38.1	(32.7–43.4)	17.1	(16.0–18.3)	<0.01	<0.001	<0.001
Frequent mental distress ^c^	55.4	(48.7–62.1)	46.7	(39.0–54.3)	30.8	(28.2–33.4)	NS	<0.001	<0.001
Frequent physical distress ^d^	58.3	(51.4–65.2)	48.6	(41.2–55.9)	33.4	(30.8–36.0)	NS	<0.001	<0.001
Difficulty walking or climbing stairs ^e^	47.7	(42.7–52.7)	32.6	(27.9–37.4)	13.0	(12.2–13.8)	<0.01	<0.01	<0.01
**Respiratory Symptoms**									
Frequent shortness of breath ^f^	29.6	(24.8–34.5)	13.0	(8.7–17.3)	14.3	(10.9–17.6)	<0.01	<0.001	NS
Frequent productive cough ^g^	37.6	(31.7–43.5)	39.5	(34.1–44.8)	3.7	(3.1–4.1)	NS	<0.001	<0.001
Dyspnea on exertion ^h^	58.7	(52.8–64.5)	55.6	(50.5–60.8)	10.1	(9.1–11.1)	NS	<0.001	<0.001

DN: Diagnosed; HR: High-Risk; LR: Low-Risk. ^a^ Diagnosed-Self-reported physician diagnosis of COPD. High-risk-No COPD diagnosis, ≥10 years of cigarette/tobacco use, and at least one respiratory symptom (frequent productive cough, frequent shortness of breath (SOB), or breathing problems limited physical activity in past year.) Low-risk-Neither diagnosed nor high-risk. ^b^ Would you say that in general your health is excellent, very good, good, fair or poor? ^c^ Number of days mental health not good in last month ≥14 days? ^d^ Number of days physical health not good in last month ≥14 days? ^e^ Do you have serious difficulty walking or climbing stairs; ^f^ Shortness of breath all of the time or most of the time in the last 30 days; ^g^ Cough up phlegm or mucus every day or most days in the last 30 days; ^h^ Agree slightly or strongly that physical activity is affected by shortness of breath over the past year.

**Table 4 healthcare-07-00012-t004:** State-specific, age-adjusted prevalence of COPD risk-categories among adults ≥18 years-old in each of the four states (FL, KY, SC, TX) using the 2015 Behavioral Risk Factor Surveillance system telephone survey.

Characteristic	DN with COPD ^b^ (n ^a^ = 3915) (%, 95% CI)		HR for COPD ^b^ (n ^a^ = 2399) (%, 95% CI)		LR for COPD ^b^ (n ^a^ = 29,585) (%, 95% CI)		Comparison of Groups (*P*-Value)		
							DN vs. HR	DN vs. LR	HR vs. LR
**Gender**									
Men	6.2	(5.4–6.9)	6.3	(5.5–7.1)	87.6	(86.5–88.6)	NS	<0.001	<0.001
Women	7.0	(6.4–7.7)	4.1	(3.5–4.9)	88.8	(87.9–89.7)	<0.001	<0.001	<0.001
**State**									
FL	6.5	(5.5–7.5)	5.6	(4.5–6.8)	87.2	(86.4–89.3)	NS	<0.001	<0.001
KY	11.2	(10.1–12.3)	9.0	(7.7–10.3)	79.8	(78.2–81.4)	NS	<0.001	<0.001
SC	7.1	(6.5–7.8)	6.1	(5.4–6.8)	86.7	(85.8–87.7)	NS	<0.001	<0.001
TX	5.7	(5.0–6.4)	3.7	(3.2–4.2)	90.5	(89.7–91.4)	<0.001	<0.001	<0.001

DN: Diagnosed; HR: High-Risk; LR: Low-Risk. ^a^ Unweighted sample size; ^b^ Diagnosed-Self-reported physician diagnosis of COPD. High-risk-No COPD diagnosis, ≥10 years of cigarette/tobacco use, and at least one respiratory symptom (frequent productive cough, frequent shortness of breath (SOB), or breathing problems limited physical activity in past year.) Low-risk-Neither diagnosed nor high-risk.

**Table 5 healthcare-07-00012-t005:** State-specific, age-adjusted prevalence of frequent shortness of breath among COPD risk categories among adults ≥45 years-old in each of the four states (FL, KY, SC, TX) using the 2015 Behavioral Risk Factor Surveillance system telephone survey.

Characteristic		Diagnosed with COPD ^b^ (n ^a^ = 3915) (%, 95% CI)		High-risk for COPD ^b^ (n ^a^ = 2399) (%, 95% CI)		Low-risk for COPD ^b^ (n ^a^ = 29,585) (%, 95% CI)	
Overall		62.4	(57.7–67.1)	16.7	(12.9–20.6)	20.8	(17.2–24.5)
State	Florida	69.0	(59.3–78.6)	17.7	(8.9–26.0)	13.3	(7.3–19.3)
	Kentucky	66.0	(60.0–72.8)	13.7	(8.4–19.0)	19.9	(15.2–24.6)
	South Carolina	60.3	(53.7–66.9)	19.3	(14.2–24.4)	20.4	(15.0–25.9)
	Texas	54.4	(45.5–62.3)	16.4	(11.3–21.4)	29.2	(21.9–36.5)

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
