# Peer review of "Use of a Cross-Sectional Survey in the Adult Population to Characterize Persons at High-Risk for Chronic Obstructive Pulmonary Disease"

_healthcare, 2019, doi:10.3390/healthcare7010012_

Round 1
Reviewer 1 Report
Dr. Roy A. Pleasants described the use of a survey to chracterize the risk of COPD.
Several comments are made below.
# Authors needs to show reference for the definition COPD risk groups (page 3, lines 96). Was there any study showing that high- or low-risk groups develop future COPD diagnosis ? What would be the definition of subjects with no-risk for COPD ?
# In figure 1, 6 lines were shown, but 3 legends (○, X, etc) were missing.
Author Response
Dr. Roy A. Pleasants described the use of a survey to chracterize the risk of COPD. Several comments are made below. #
Authors needs to show reference for the definition COPD risk groups
(page 3, lines 96). Was there any study showing that high- or low-risk
groups develop future COPD diagnosis ? What would be the definition of
subjects with no-risk for COPD ? Response: The choice of high-risk vs low-risk was based on prior study (cited -
Pleasants in 2012). GOLG Guidelines recommend considering screening
persons with resp sxs and exposure history (e.g. > 10 years smoking),
thus our study methods definitions for high vs low-risk for COPD are
compatible with GOLD guidelines. # In figure 1, 6 lines were shown, but 3 legends (○, X, etc) were missing.
Response: The Legend for Figure 1 I submitted should have 6 descriptors, not sure how it got 'cut'off". Now they are all correct.
Reviewer 2 Report
The study by Pleasants et al., characterizes at high risk COPD patients based on a cross sectional study utilizing the enchanced BRFSS questionnaire in 4 different states of the US. The main findings of this work include correlation of high risk factors to COPD disease incidence using the modified BRFSS survey system. However, the authors consider only cigarettes smoking as a primary risk factor (which indeed it is) but does not cover other known risk factors for COPD (asthma, chronic bronchitis, pollution, AAT deficiency etc.,) But since this study was conducted using the predefined BRFSS survey, these additional risk factors can be elaborated as caveats to this survey system in the discussion.
As the results in this paper are based on a modified BRFSS survey questionnaire, the work is novel as I do not see any other publications reporting COPD using the modified questionnaire.
The claims are convincing as the authors use well defined algorithms and statistics and the results are a direct extrapolation of the survey data so no further analysis seem required. Future work can be reported as a followup work as this survey has already been conducted using the standard set of questionnaires. So, any additional evaluations may literally require a new study altogether. So, the work in the present form suffices the proposed rationale.
Background part can be further elaborated in the context of previous literature and the implications of this work.
Author Response
The
study by Pleasants et al., characterizes at high risk COPD patients
based on a cross sectional study utilizing the enchanced BRFSS
questionnaire in 4 different states of the US. The main findings of this
work include correlation of high risk factors to COPD disease incidence
using the modified BRFSS survey system. However, the authors consider
only cigarettes smoking as a primary risk factor (which indeed it is)
but does not cover other known risk factors for COPD (asthma, chronic
bronchitis, pollution, AAT deficiency etc.,) But since this study was
conducted using the predefined BRFSS survey, these additional risk
factors can be elaborated as caveats to this survey system in the
discussion.
Response: In the Discussion (limitations) , we point out we did not address non-smoking etiologies in the high or low risk groups.
As the results in this paper are based on a modified
BRFSS survey questionnaire, the work is novel as I do not see any other
publications reporting COPD using the modified questionnaire.
The
claims are convincing as the authors use well defined algorithms and
statistics and the results are a direct extrapolation of the survey data
so no further analysis seem required. Future work can be reported as a
followup work as this survey has already been conducted using the
standard set of questionnaires. So, any additional evaluations may
literally require a new study altogether. So, the work in the present
form suffices the proposed rationale.
Background part can be further elaborated in the context of previous literature and the implications of this work.
Response: WE addressed background and implications extensively in the Discussion and added more detail in Introduction about value of addressing persons wat high risk for COPD.
Reviewer 3 Report
In the manuscript of Pleasants et al., the authors used the BRFSS health survey to describe the epidemiology of self-reported COPD and patients at risk for COPD in the US. In accordance to their previous publication in the Journal of COPD in 2015, they followed the same algorithm including statistical analysis but instead of 1 state, included now 4 states. They specifically show that there is a younger population (25-34 years) that is at increased risk for COPD per their definition. Furthermore, duration of tobacco use was associated with increased prevalence of COPD.
In summary, the authors conclude that the data obtained from the modified BRFSS will help with current CDC initiatives to screen for lung health in order to help prevent disease.
Overall, I think that the manuscript is well written, but there is lack of novelty, since the authors mainly apply their recent analysis from 1 state to 4 states now. This though is important, since it will be more representative for the US population. There are some minor concerns, which should be addressed.
Minor concerns:
1. Most evaluation of COPD and COPD risk is based on cigarette smoking: pollution and childhood asthma are not included as important factors
2. Page 1 line 38/39 is too speculative. Since the authors only screened for COPD, the modified BRFSS will not necessarily define lung health.
3. Page 2, line 77-81, please change font size.
4. The definition of the COPD risk groups could also be a risk group for other smoking related lung disease and are not specific for COPD.
5. In the results, the symptom questionnaire in FL was only aksed over 6 months – was that taken into account for statistical analysis?
6. Table 2: how representative is the analyzed group when compared to the whole US? This should be included in the discussion.
7. Fig. 1 and 2 diagrams should not have connecting lines, since these are not continuous variables.
8. Line 227: I would avoid to mention “general lung health”, since this questionnaire was tailored towards COPD only, which is only one lung disease.
Author Response
In the manuscript of Pleasants et al., the authors used the BRFSS health survey to describe the epidemiology of self-reported COPD and patients at risk for COPD in the US. In accordance to their previous publication in the Journal of COPD in 2015, they followed the same algorithm including statistical analysis but instead of 1 state, included now 4 states. They specifically show that there is a younger population (25-34 years) that is at increased risk for COPD per their definition. Furthermore, duration of tobacco use was associated with increased prevalence of COPD.
In
summary, the authors conclude that the data obtained from the modified
BRFSS will help with current CDC initiatives to screen for lung health
in order to help prevent disease.
Overall, I think that the manuscript is well written, but there is lack of novelty, since the authors mainly apply their recent analysis from 1 state to 4 states now. This though is important, since it will be more representative for the US population. There are some minor concerns, which should be addressed.
Response: Thank you very much.
Minor concerns:
1. Most evaluation of COPD and COPD risk is based on cigarette smoking: pollution and childhood asthma are not included as important factors.
Response: Non-smoking risk for COPD addressed in Discussion.
2. Page 1 line 38/39 is too speculative. Since the authors only screened for COPD, the modified BRFSS will not necessarily define lung health.
Response: As the survey addresses resp sxs, clearly the survey does address lung health. We are preparing a separate manuscript describing resp sxs in the gen population using this data. I would consider COPD a major aspect of lung health.
3. Page 2, line 77-81, please change font size.
Response: Done.
4. The definition of the COPD risk groups could also be a risk group for other smoking related lung disease and are not specific for COPD.
Response: We did not address other types of tobacco induced lung disease as lung cancer, nor did we imply that we did.
5. In the results, the symptom questionnaire in FL was only aksed over 6 months – was that taken into account for statistical analysis?
Response: Yes, we consider the differences in 6 months vs 12 month of data from Florida.
6. Table 2: how representative is the analyzed group when compared to the whole US? This should be included in the discussion.
Response: We included a sentence in the Limitations.
7. Fig. 1 and 2 diagrams should not have connecting lines, since these are not continuous variables.
Response: Editor to change.
8. Line 227: I would avoid to mention “general lung health”, since this questionnaire was tailored towards COPD only, which is only one lung disease.
Response: Although the questions help identify COPD, they also help define lung health based on the most common resp sxs in the US.
Round 2
Reviewer 1 Report
The reply was sufficient.